# Association between TSH levels within the reference range and adiposity markers at the baseline of the ELSA–Brasil study

**Maria de Fátima Haueisen Sander Diniz**[1]*, **Alline Maria Rezende Beleigoli**[1,2], **Isabela M. Benseñor**[3,4], **Paulo A. Lotufo**[3,4], **Alessandra C. Goulart**[4], **Sandhi Maria Barreto**[1]

**1** Universidade Federal de Minas Gerais, Belo Horizonte, Brazil, **2** Flinders University, Adelaide, Australia, **3** Universidade de São Paulo, São Paulo, Brazil, **4** Hospital Universitário, Universidade de São Paulo (HU-USP), São Paulo, Brazil

* mfhsdiniz@yahoo.com.br

**Data Availability Statement:** Due to ethical restrictions approved by the ethics committee of each institution (Universidade Federal de Minas

## Abstract

### Background

The association of thyrotropin (TSH) with overall (body mass index, BMI), visceral (waist circumference and steatosis), and upper subcutaneous (neck circumference, NC) adiposity markers is still controversial, and the aim of this study is to assess these associations in the baseline data of a large cohort from ELSA-Brasil.

### Methods and findings

This cross-sectional study included 11,224 participants with normal thyroid function (normal TSH levels). BMI, waist circumference, NC and steatosis, defined by hepatic attenuation (mild or moderate/severe) were the explicative variables. TSH levels were log transformed (logTSH), and multivariate linear regression models were generated to estimate the associations between logTSH and BMI (continuous and categorized), waist circumference, NC, and steatosis after adjusting for sociodemographic characteristics, health behaviors, and comorbidities. The mean age was 51.5±8.9 years, 5,793 (51.6%) participants were women, 21.8% (n = 2,444) were obese, and 15.1% of the sample was TPOAb positive. The TSH levels were significantly higher in the obese group than in the reference group (<25.0 kg/m$^2$). In the multivariable linear regression models, significant associations of logTSH with BMI and obesity were found. LogTSH was associated with waist circumference only among women. NC and steatosis were not related to TSH levels.

### Conclusions

TSH levels were associated with overall adiposity and obesity. Further studies may elucidate reference levels of TSH according to BMI status.

Gerais, Universidade de São Paulo, Universidade Federal do Espírito Santo, Universidade Federal do Rio Grande do Sul, Universidade Federal da Bahia e Fundação Oswaldo Cruz) and by the Publications Committee of ELSA-Brasil (publiELSA), the data used in this study can be made available for research proposals by a request to ELSA's Datacenter (rb.sgrfu@asleacitsitatse) and to the ELSA's Publications Committee. Additional information can be obtained from the ELSA Coordinator from the Research Center of Minas Gerais (sandhi.barreto@gmail.com).

**Funding:** Funding: The ELSA-Brasil study was supported by the Brazilian Ministry of Health (Science and Technology Department- DECIT) and the Brazilian Ministry of Science and Technology (Financiadora de Estudos e Projetos and CNPq National Research Council) (grants 01 060010.00 RS, 01 06 0212.00 BA, 01 06 0300.00 ES, 01 06 0278.00 Mg, 01 06 0115.00 SP, 01 06 0071.00 RJ).". Grants are for ELSA-Brasil study, not for any individual author.

**Competing interests:** The authors have declared that no competing interests exist.

## Introduction

Thyrotropin (TSH) is a well-recognized pituitary hormone that binds to its receptor on the thyroid gland, promoting thyroid function. However, TSH receptors (TSHR) are also expressed in many other tissues, including adipose tissue. TSHR activation has been linked to white adipose tissue lipolysis in animal and human cell models[1] and has been associated with triglyceride accumulation in animal models.[2] TSHR stimulation is also involved in the regulation of thermogenesis, and thyroid hormones might contribute to the control of energy expenditure and metabolic rate.[3] Thus, the effects of TSHR activation on adipose tissue may potentially impact body composition.[4]

Thyroid dysfunctions, namely, hyper- or hypothyroidism, are often associated with weight changes. Conversely, treatment of subclinical hypothyroidism does not seem to be beneficial for weight loss purposes.[5] A systematic review that included 29 studies showed an association between TSH and adiposity markers, such as body mass index (BMI) and waist circumference among individuals with normal thyroid function in 18 studies,[6] even independently of free thyroxine levels.[7] However, among these studies, some adjusted for smoking status[7–17] and others do not. [18–21] Also, the upper reference limit of TSH was not the same among the studies.[7–21] Interestingly, a data analysis of 14 cohorts with 55,412 individuals with thyroid function within the normal range demonstrated that BMI was not different among the lower (0.45–1.49 mIU/L) up to the higher TSH quartile (3.50–4.49 mIU/L). This analysis included studies from Europe, United States, Australia, Asia and South America.[22] Data of 16,902 participants with serum TSH within the reference range from five population-based studies from Germany, Denmark and Netherlands showed the association of TSH with BMI and waist circumference in cross-sectional analysis. However, longitudinal analyses of the four prospective studies included showed that higher TSH levels at baseline studies were related to a decrease of BMI and waist circumference.[23]

Body mass index has been considered the best marker of overall obesity and is mainly used in large epidemiological studies.[24–26] Waist circumference and steatosis are proxies of visceral adiposity.[27] These adipose deposits are metabolically different from subcutaneous deposits. In the Framingham Heart Cohort, neck circumference (NC), a measure of subcutaneous fat, has been independently associated with adverse cardiometabolic risk factors, perhaps acting as a source of circulating free fatty acids.[28] The influence of TSH on visceral adipose tissue and upper body subcutaneous fat remains to be elucidated.

In light of these points, our aim is to assess the association of TSH with overall (BMI), visceral (waist circumference and steatosis), and upper subcutaneous (neck circumference) adiposity markers in the baseline data of participants with normal thyroid function from the Brazilian Longitudinal Study of Adult Health (Estudo Longitudinal da Saúde do Adulto–ELSA-Brasil), a large Brazilian cohort in progress.

## Methods

### Study design and population

This cross-sectional analysis is a subproject of the Brazilian Longitudinal Study of Adult Health (ELSA–Brasil), which has been described previously.[29,30] Briefly, the baseline cohort comprises 15,105 active or retired civil servants of universities or research institutions from six cities in Brazil who were enrolled between August 2008 and December 2010; the participants were aged from 35–74 years and were mostly female (54%) middle-aged (78% aged < 60 years) adults. All participants were volunteers and signed an informed consent form. All the six Institutional Review Boards (Universidade Federal de Minas Gerais, Universidade de São

Paulo, Universidade Federal do Espírito Santo, Universidade Federal do Rio Grande do Sul, Universidade Federal da Bahia e Fundação Oswaldo Cruz) approved this study and also the National Commission for Ethics in Research (CONEP / MS 976/2006) This analysis was approved by the Publications Committee of ELSA-Brasil (PubliELSA 17_0567). The quality and control of the collection and storage of data were ensured via training sessions, certifications, and renewal of certifications of those performing the interviews, clinical examinations and laboratory tests involved in the study protocol.[30]

We excluded participants who were using levothyroxine (n = 1,072) or antithyroid agents (thiamazole or propylthiouracil n = 20), those with missing values for TSH (n = 11), and those with TSH levels <0.40 mcUI/mL (n = 224) or >4.0 mcUI/mL (n = 877). We also excluded participants who were using any drugs that can interfere with TSH laboratory assays (amiodarone, alfa interferon, beta interferon, bromocriptine, carbamazepine, carbidopa, phenytoin, furosemide, haloperidol, heparin, levodopa, lithium carbonate, metoclopramide, valproic acid, oxcarbazepine, propranolol, phenobarbital, primidone, and rifampicin, n = 508),[31] amfepramone (n = 10), mazindol (n = 1), or orlistat (n = 13); participants who previously underwent bariatric surgery (n = 96); and patients with self-reported cirrhosis/hepatitis (n = 1,049).

After the above mentioned participants were excluded, 11,224 participants remained in the analysis. For the steatosis analysis, the quality of the ultrasound images of 3,231 participants was unacceptable, and these exams were excluded. The final sample for steatosis was 8,993.

## Laboratory measurements

Blood samples were collected after a 12-hour fasting.[30] TSH was assayed for all participants by an immunoenzymatic bead-based technique using Siemens reagent L2KTS2 with an analytical sensitivity of 0.004 mU/L (Siemens IMMULITE 2000 Immunoassay System® – Siemens Healthcare Diagnostics, Deerfield, IL, USA). The presence of thyroid peroxidase antibody (TPOAb) was tested by electrochemiluminescence (Immulite, Los Angeles, CA, USA) and was defined as positive if the result was ≥ 35 UI/mL. All samples were stored frozen at -80˚C until the date of transportation to the Central Laboratory. All laboratory analyses were performed at a single research center (University of São Paulo).[32,33] The cutoff values of TSH were similar to those used in the National Health and Nutritional Examination Survey (NHANES III).[34]

Other laboratory tests were performed, and the methodology and equipment utilized were as follows: fasting plasma glucose and total cholesterol (enzymatic colorimetric method), HDL cholesterol (homogeneous colorimetric method without precipitation), triglycerides (glycerol phosphate peroxidase). Glycated hemoglobin A1c was assessed using high-pressure chromatography (Bio-Rad Laboratories, Hercules, CA, USA)[32]. Intra-assay coefficients of variation (CV) of the TSH, HDL cholesterol, triglycerides and A1c were, respectively, 3.32%, 2.0%,1.6%, 0.86%. Inter-assay CV for lower and higher control of TSH were 4.65 and 5.89, respectively. [33]

## Anthropometric data

Anthropometric parameters were measured using standardized and calibrated instruments according to the study protocol. Weight (kg) and height (cm) were measured while the participant was barefoot, wearing light clothing and standing straight with the head level using Toledo scales (to the nearest 100 g) and a stadiometer (accuracy of 0.1 cm), respectively. Waist circumference was measured by inelastic tapes (cm) and used as a proxy of visceral adiposity. [29] The average of two measures was used for analyses. Since BMI [weight(kg)/height(m)$^2$] is considered the most practical measure of overall obesity for populations, it was used as a proxy of "overall obesity" in this study[24–26,35]. BMI was categorized according to the World

Health Organization criteria: 18.5–24.9 kg/m$^2$—normal weight (reference category in analyses with BMI as a categorical variable), 25–29.9 kg/m2 –overweight and $\geq$ 30kg/m$^2$- obesity.[35] Neck circumference was measured with an inelastic tape (mm) right above the cricoid cartilage and perpendicular to the long axis of the neck, with the participant in a sitting position.[29] It was used as a proxy of subcutaneous adipose tissue in our study.

## Hepatic ultrasound protocol and steatosis

Liver ultrasound examinations were performed on the participants by board-certified radiologists or by radiology technicians after adequate training using the same models of equipment: a high-resolution B-mode scanner (SSA-790A, Aplio XG, Toshiba Medical System, Tokyo, Japan) and a convex array transducer (model PVT-375BT), with a central frequency of 3.5 MHz and a fundamental frequency of 1.9–5.0 MHz. After the acquisition process, the B-mode hepatic ultrasound images were read by board-certified radiologists at the ELSA-Brasil site, which was established as the ELSA-Brasil ultrasound reading center. Hepatic attenuation was classified as normal or abnormal as follows: mild, moderate or severe.[36] Steatosis was defined by hepatic attenuation (mild or moderate/severe) and was used as a proxy of abdominal fat.

**Other participant characteristics.** Baseline data collection (standardized interviews, anthropometric and blood pressure measurements, blood tests) was performed at the workplace and at the six research centers. Race/skin color (white, black, or mixed: brown, Asian, native) and educational level [$\geq$12 years (university degree); 9–11 years (high school); $\leq$8 years (elementary school)] were self-reported. Participants were classified as nonsmokers or as current smokers if they had smoked at least 100 cigarettes (five packs of cigarettes) throughout their lifetime and currently smoked. Alcohol use was categorized as low (weekly consumption $\leq$ 175 g), moderate (176–350 g) or high ($>$350 g).

Leisure and transportation-related physical activity was verified through the long form of the International Physical Activity Questionnaire (IPAQ), and participants were categorized into three groups: low, moderate or high activity, according to the sum of the metabolic equivalents per week (combination of activity type, frequency and duration). Diabetes mellitus was defined by insulin and antidiabetic drug use or by a self-reported medical diagnosis of diabetes. Cardiovascular disease was defined by the combination of self-reported heart failure, cardiac revascularization, stroke or previous myocardial infarction.[10] The glomerular filtration rate was estimated by the Chronic Kidney Disease Epidemiology Collaboration (CKD-EPI) equation.[37]

## Statistical analysis

The normality of the distribution of the data was assessed by histograms, kurtosis and Shapiro-Wilk tests. TSH (response variable) had a skewed distribution and was log-transformed (logTSH). Descriptive and univariate analyses were used to compare logTSH with BMI (continuous and categorized), waist circumference, NC, and steatosis (explicative variables); sociodemographic characteristics; health behaviors; and comorbidities. Multivariate linear regression models were generated to estimate the associations between logTSH and (1) BMI (continuous or categorized as 18.5–25.0 kg/m$^2$ (reference), $\geq$ 25–29.9 kg/m$^2$, and $\geq$ 30 kg/m$^2$), (2) waist circumference (continuous by sex), (3) NC (continuous by sex), and (4) hepatic steatosis (no, mild, moderate/severe). All models were adjusted for the following covariates: Model 1- age (years) and sex (only for BMI and steatosis); Model 2- Model 1 plus race/skin color (white, black or mixed), education, TPOAb (negative/positive), diabetes (yes/no), cardiovascular disease (yes/no), and glomerular filtration rate; and Model 3 –Model 2 plus lifestyle

behaviors (smoking, alcohol use, physical activity). We tested whether BMI and diabetes, BMI and smoking and BMI and TPOAb were effect modifiers on the associations of LogTSH by multiplicative interaction terms between BMI/diabetes, BMI/smoking and BMI/ TPOAb. Sensitivity analyses according to TPOAb status (negative/positive) and smoking status (current smoker/nonsmoker) were performed when the p value of the interaction term was < .05. All p values given are two-sided with the level of significance set to p < .05. We used the STATA™ package 14.0 (Stata Corporation, College Station, TX, USA) for all analyses.

## Results

Among the 11,224 participants included in the present analysis, the mean age was 51.5±8.9 years, and almost fifty percent were female and had a university degree. Twenty-two percent (n = 2,444) of the sample had obesity, and 40.5% (n = 4,548) were overweight. The median TSH level was 1.48 mcUI/mL [interquartile range (IQR 1.04–2.18], and 15.1% of the sample was TPOAb positive. Among BMI categories, TSH levels were significantly higher in the obesity group than in the reference group (<25.0 kg/m$^2$), with median values of 1.53 mcUI/mL [Interquartile range (IQR) 1.07–2.23] and 1.47 mcUI/mL [IQR1.04–2.12], respectively. TSH levels were not different between the overweight and normal weight groups (p = 0.18) or the overweight and obesity groups (p = 0.32). The characteristics of the study population are depicted in Table 1.

In univariate analysis, logTSH was associated with BMI and waist circumference. LogTSH was not associated with neck circumference or steatosis. The association between TSH levels and BMI categorized as reference, ≥ 25–29.9 kg/m$^2$, and ≥ 30 kg/m$^2$ was only significant for the obesity category.

In multivariable linear regression models, a significant association was found between logTSH and BMI. The final model also included sex, skin color/race, glomerular filtration rate, TPOAb positivity and smoking. As thyrotropin levels were log-transformed, for each 1 kg/m$^2$

**Table 1. Demographic, lifestyle characteristics, thyrotropin, anthropometric data, and comorbidities of study population: Brazilian Longitudinal Study of Adult Health (2008–2010) n = 11,224.**

| Characteristic * | n (%) |
| --- | --- |
| **Sex** | |
| Male | 5,440(48.5) |
| Female | 5,784(51.5) |
| **Age**(years) | 51.5±8.9* |
| **Skin color/race**** | |
| White | 5,540 (50.0) |
| Black | 1,891 (17.0) |
| Mixed | 3,658 (33.0) |
| **Educational level** | |
| Elementary school | 1,462(13.0) |
| High school | 4,020(35.8) |
| University degree | 5,742 (51.2) |
| **TSH** (mcUI/mL) | 1.48[1.04–2.18]$^\infty$ |
| **TPOAb positive** | 1,690(15.1) |
| **Smoking status** | |
| No | 9,719(86.6) |
| Yes | 1,505(13.4) |

*(Continued)*

**Table 1.** (Continued)

| Characteristic * | n (%) |
|---|---|
| **Alcohol Consumption**** | |
| <175g/week | 10,152(91.0) |
| 175-349g/week | 775(6.4) |
| ≥ 350g/week | 285(2.6) |
| **Leisure physical activity**** | |
| Highly active | 793(7.1) |
| Moderate active | 1,759(15.9) |
| Low active | 8,524(77.0) |
| **BMI** (kg/m$^2$) | 26.9 ± 4.6* |
| **Waist circumference** (cm) | |
| Male | 95.0 ± 11.5* |
| Female | 87.1 ± 12.4* |
| **Neck circumference** (cm) | |
| Male | 39.5 ± 2.8* |
| Female | 33.9 ± 2.5* |
| **Hepatic attenuation**$^§$ | |
| Normal | 5,453 (60.6) |
| Mild | 3,267 (36.4) |
| Moderate/severe | 273(3.0) |
| **Comorbidities and exams** | |
| Diabetes mellitus (Yes) | 1,005 (8.9) |
| CVD (Yes) | 521(4.6) |
| GFR(mL/min/1.73 m$^2$) | 97.0[94.5–118.1] $^∞$ |
| Fasting plasma glucose(mg/dL) | 100.2[94,2–108.8] $^∞$ |
| Glycated hemoglobin A1c(%) | 5.3[4.9–5.8] $^∞$ |
| Total cholesterol(mg/dL) | 212[186–239] $^∞$ |
| HDLc(mg/dL) | 54[46–64] $^∞$ |
| Tryglicerides(mg/dL) | 114[81–165] $^∞$ |

$^∞$Median [Interquartile Range]

* Mean± standard deviation; Skin color- self-reported

Mixed- brown, Asian, or native; TSH- Thyrotropin; TPOAb- Thyroid peroxidase antibody; BMI- body mass index;

CVD- cardiovascular disease; GFR- glomerular filtration rate; HDLc- HDL cholesterol.

**Differences in total frequency (n) of characteristics are due to missing values.

$^§$ Final sample for steatosis n = 8,993.

increase in BMI, TSH increased by 0.4%. LogTSH was associated with waist circumference only among women (β exponential ($e^β$) = 1.001[1.0006–1.003] p<0.05), and the final model included skin color/race, education, TPOAb positivity, alcohol use and smoking status. LogTSH was not associated with neck circumference or steatosis (Table 2).

The final model of the association between logTSH and categorized BMI was significant only for the obesity category and included sex, race/skin color, glomerular filtration rate, TPOAb positivity, and smoking (Table 3).

The association between LogTSH and adiposity markers were not influenced by diabetes (p value for the interaction terms = 0.47 for BMI, 0.99 for categorized BMI, 0.99 for waist circumference, 0.99 for NC and 0.97 for steatosis). The association of LogTSH with BMI, categorized BMI, waist and neck circumference were not influenced by smoking (p value for the

**Table 2. Linear regression to estimate the association between thyrotropin[1] and body mass index, waist circumference, neck circumference, and hepatic steatosis.** Brazilian Longitudinal Study of Adult Health (2008–2010) n = 11,224.

| | Crude | Final Model |
|---|---|---|
| | $e^{\beta}$ [CI95%] | $e^{\beta}$ [CI95%] |
| **BMI**(kg/m$^2$) | 1.004[1.002–1.006]* | 1.004[1.003–1.007]* |
| **Waist**(cm) | | |
| Male | 1.001[1.0002–1.002]* | 1.000[0.999–1.002] $^\infty$ |
| Female | 1.001[0.99–1.002] $^\infty$ | 1.001[1.0006–1.003]* |
| **Neck**(cm) | | |
| Male | 1.004[0.99–1.008] $^\infty$ | 1.002[0.998–1.007] $^\infty$ |
| Female | 0.999[0.99–1.005] $^\infty$ | 1.004[0.999–1.010] $^\infty$ |
| **Steatosis**$^\$$ | | |
| No | Reference | Reference |
| Mild | 0.993[0.97–1.014] $^\infty$ | 0.991[0.970–1.012] $^\infty$ |
| Moderate/Severe | 1.019[0.96–1.083] $^\infty$ | 1.005[0.946–1.067] $^\infty$ |

Thyrotropin[1]- Log TSH; e$\beta$ – $\beta$ exponential; [CI95%]–confidence interval 95%

BMI- body mass index

*p value < .05

$^\infty$ p value> .05

$^\$$ For steatosis analysis n = 8,993.

interaction terms were not significant). However, the interaction term significance was borderline for the association LogTSH and waist among women (p = 0.08), and for moderate/severe steatosis (p = 0.05). The association of LogTSH with BMI, categorized BMI, and waist circumference were not influenced by TPOAb (p value for the interaction terms were not significant). However, the association between LogTSH and neck circumference among men was influenced by TPOAb (p value for the interaction = 0.04). The association of LogTSH and moderate/severe steatosis was also influenced by TPOAb (p value for the interaction = 0.01).

After stratification by smoking status, LogTSH remained associated with waist circumference only among women, and there was no association with steatosis (Table 4).

After analyses stratified by TPOAb, in the TPOAb-positive group, LogTSH was associated neck circumference among men and with moderate/severe steatosis (Table 5).

**Table 3. Linear regression to estimate the association between thyrothropin[1] and body mass index status.** Brazilian Longitudinal Study of Adult Health (baseline 2008–2010) n = 11,224.

| | Crude | Final Model |
|---|---|---|
| | $e^{\beta}$ [CI95%] | $e^{\beta}$ [CI95%] |
| **BMI status** | | |
| Normal weight | Reference | Reference |
| Overweight | 1.01[0.99–1.03] $^\infty$ | 1.02[0.99–1.04] $^\infty$ |
| Obesity | 1.04[1.01–1.07] * | 1.05[1.02–1.08]* |

Thyrothropin[1]-logTSH; e$\beta$ – $\beta$ exponential; [CI95%]–confidence interval 95%; BMI-

body mass index; Normal weight- BMI <25.0 kg/m$^2$; Overweight- BMI ≥25–29.9 kg/m$^2$

Obesity- BMI ≥30.0 kg/m$^2$

* p value < .05

$^\infty$ p value> .05

**Table 4. Association between Thyrotropin[1] and waist, and with steatosis after stratification by smoking.** Brazilian Longitudinal Study of Adult Health (2008–2010).

| | Smokers Final Model $e^{\beta}$ [CI95%] | Non smokers Final Model $e^{\beta}$ [CI95%] |
|---|---|---|
| **Waist[2]** | | |
| Men | 1.000[0.997–1.003] [§] | 1.000[0.999–1.002] [§] |
| Women | 1.003[1.0001–1.006]* | 1.001[1.0002–1.002]* |
| **Steatosis[3]** | | |
| No | Reference | Reference |
| Mild | 0.996[0.938–1.056] [§] | 0.992[0.970–1.016] [§] |
| Moderate/Severe | 0.791[0.624–1.002] [§] | 1.027[0.965–1.093] [§] |

[1]Thyrotropin- Log TSH

[2]n = n = 11,224

[3]n = 8,993; $e\beta$ – $\beta$ exponential; [CI95%]–confidence interval 95%

Waist- waist circumference

* p value < .05

[§]p value>.05

## Discussion

Among the baseline participants of the ELSA-Brasil who had serum TSH concentrations within the reference range, higher TSH levels were associated with a higher BMI after all adjustments for confounders. In the analysis with BMI categories, the association of higher TSH levels was present only among participants with a BMI$\geq$ 30 kg/m$^2$, in comparison with normal weight and overweight categories. Interestingly, TSH levels were also associated with waist circumference only among women. TSH levels were associated neither with steatosis, which is a proxy of visceral adiposity, nor with neck circumference, an upper body subcutaneous adipose tissue deposit that is a predictor of cardiovascular diseases.[28] Similar to our study, several previous studies have found a direct association between TSH and BMI, irrespective of sex.[7–11] In samples composed of only women[38,39] or men,[18] the TSH-BMI association was also demonstrated. However, some authors reported no association[12–14,19,40] or associations only among men[41] or among women.[15] These divergences could

**Table 5. Association between thyrothropin[1], waist and neck circumference, and steatosis after stratification by thyroid peroxidase antibody.** Brazilian Longitudinal Study of Adult Health (2008–2010).

| | TPOAb positive Final Model[1] $e^{\beta}$ [CI95%] | TPOAb negative Final Model[1] $e^{\beta}$ [CI95%] |
|---|---|---|
| **Neck** | | |
| Men | 1.015[1.003–1.029]* | 0.999[0.995–1.005] [§] |
| Women | 1.006[0.994–1.018] [§] | 1.004[0.997–1.009] [§] |
| **Steatosis[2]** | | |
| No | | |
| Mild | Reference | Reference |
| Moderate/Severe | 1.025[0.963–1.089] [§] | 0.985[0.963–1.008] [§] |
| | 1.198[1.016–1.413]* | 0.981[0.919–1.046] [§] |

Thyrothropin[1]- logTSH; $e\beta$ – $\beta$ exponential; [CI95%]–confidence interval 95%

Neck- neck circumference; Waist- waist circumference; TPOAb—Thyroid peroxidase antibody.

*p value < .05

[§]p value>.05. For neck analysis-n = 11,224 [2] For steatosis analysis, n = 8,993.

partially result from diverse methodological aspects among both cross-sectional and longitudinal studies with regard to exclusion criteria, model adjustments, stratification, BMI levels of the groups, and even TSH cut-offs. Although clinical or subclinical hypo- or hyperthyroidism, previous thyroid disorder, or use of medication related to a thyroid disorder were included in the exclusion criteria in these studies, only some of the studies mentioned the exclusion of drugs that interfere with thyroid function.[7,39]

Adipocytes may play a role in TSH physiologic regulation. Whether this role is mediated by insulin resistance or is related to the adipose tissue accumulation per se is still a controversy among authors. Nannipieri et al. found a reduction in TSHR expression in the subcutaneous and visceral adipose tissue of people with severe obesity compared to the expression in the adipose tissue in lean individuals. After major weight loss induced by bariatric surgery, TSHR expression increases significantly in subcutaneous adipose tissue, and there was a reduction in circulating TSH and free triiodothyronine levels.[42] Fontenelle et al. noted that adiposity accumulation results in an adaptive processes to increase energy expenditure, such as changes in the activity of deiodinases, which could be associated with alterations in TSH levels.[4] With the increase of adipose tissue, the resulting hyperinsulinemia and insulin resistance decreases activity of type 2 deiodinase in thyrotrophic cells, reducing intracellular availability of triiodothyronine. This resembles a "state of hypothyroidism" and may increase TSH levels, which might contribute to explain the higher levels of TSH in individuals with obesity. However, the relationship between TSH and BMI may be independent of insulin resistance, according to some studies.[43,44] Since diabetes—a condition associated with insulin resistance- was not a modifier of the association between BMI and TSH, our findings suggest that the association TSH-BMI is independent of insulin resistance.

We found a direct association between TSH and waist circumference, only among women, in the overall population. This result is different from the findings of HUNT 2 and 3, in which an increase in TSH was associated with an increased waist circumference among euthyroid women and men over a 10.5-year follow-up.[16] The result also differs from the 2007–2008 analysis of the National Health and Nutrition Examination Survey (NHANES), in which every 1-quartile increase in waist circumference was associated with an increase in TSH for women and men.[7] TSH levels were not associated with waist circumference among Japanese men or women.[41] A cross-sectional analysis that used data from four population-based cohort studies and one population-based cross-sectional study from Germany, the Netherlands and Denmark found that serum TSH within the reference range was positively associated with waist circumference.[23] Conversely, our findings are similar to those reported by DePergola et al., who found a positive weak correlation between TSH and waist circumference (r = 0.17) only among obese or overweight women.[19] Also, at a cross-sectional analysis of Study of Health in Pomerania the association of waist circumference and BMI and TSH were found only in women.[45] Two other reviews included cross-sectional studies that found an association between TSH and waist circumference irrespective of sex, among people with overweight or obesity or in a large sample of women.[4,6] It is unclear whether these findings might be related to sex differences in subcutaneous abdominal and visceral tissue proportions because women had proportionally higher fat mass than men. Camhi et al. found that, for a given level of waist circumference or BMI, women had higher levels of subcutaneous fat than men,[46] and as demonstrated by Nannipieri et al., TSHR expression is reduced in adipose tissue.[42] Perhaps more accurate measures of visceral and subcutaneous adipose tissue, as Dual-energy X-ray absorptiometry or computed tomography, could contribute to explaining the different results between studies.

Similar to a meta-analysis with 26,147 participants in which no association of nonalcoholic fat liver disease with TSH levels was found,[47] we did not find a significant association

between TSH and hepatic steatosis. Moreover, a cross-sectional analysis of a population-based Study of Health in Pomerania Trend (SHIP Trend) found no association between visceral adipose tissue, measured using magnetic resonance imaging, and TSH levels.[48]

According to some authors, current smoking may modify the association between TSH and BMI.[11,12,41,49] However, in this ELSA-Brasil study, associations of TSH with waist and steatosis remained the same after stratification by smoking status. It is still controversial whether the presence of TPOAb can modify the TSH-BMI association. In present study, the presence of TPOAb modified the association of TSH with NC among men and the association of TSH with steatosis only within the moderate to severe group. In a Brazilian study of women with normal TSH values, BMI was significantly related to TSH only among the TPOAb-positive group.[49] However, in a sample of nonobese Chinese individuals with normal thyroid function, TPOAb was not correlated with BMI,[50] and after a 10-year follow-up of 1,100 euthyroid persons, no association was observed between change in TSH and TPOAb status and weight change.[14]

Despite the significant associations between TSH and BMI and between TSH and waist circumference at the baseline of the ELSA-Brasil study, the clinical relevance of these associations is not well understood. We wonder whether other individual factors have a stronger influence on this association. Moreover, the clinical meaning of this association needs to be more thoroughly investigated in longitudinal studies, particularly regarding whether TSH reference levels should change in accordance with BMI. In the HUNT2 Study, the increase in TSH was related to a discrete increase in weight and BMI after 10.5 years in both sexes.[16] In an Iranian population, changes in TSH levels over 10 years were associated neither with the incidence of metabolic syndrome nor with any of its components, such as obesity or waist circumference. [14] In the DanThyr Study, TSH levels were not a determinant of weight change after 11 years of follow-up, despite the presence of an association between weight change and change in TSH levels.[13] Contrasting with a positive cross-sectional association demonstrated between TSH and BMI and waist circumference for women and men, Tiller at al. found that high TSH serum levels were associated with decreased anthropometric markers at longitudinal data.[45]

The large sample of adults from a middle-income country, the methodological rigor used in data collection, the centralized analysis of the laboratory tests, the quality assurance control, the exclusion of drugs that interfere with thyroid function, the large set of covariates and the stratified analysis for smoking and TPOAb status are strengths of this ELSA-Brasil study. Few studies have investigated the association of TSH and other adipose depots, such as neck circumference, waist circumference and steatosis. The limitations of this study are the absence of free thyroxine data, the lack of body composition data that prevents us from better assessing the impact of TSH on lean or fat mass, and the use of hepatic ultrasound, which might have limited the accuracy of the detection of mild steatosis.[27]

In conclusion, TSH levels were associated with overall adiposity and obesity in this cross-sectional analysis of the ELSA-Brazil cohort study. On the other hand, the visceral adiposity markers, steatosis and waist circumference, had no association or only an association among females, respectively. Upper adiposity deposits, represented by neck circumference, were not related to TSH levels. Smoking status and TPOAb presence were relevant but did not modify the main associations found. This interesting topic remains controversial and deserves further study, especially the reference levels of TSH according to BMI status.

## Acknowledgments

The authors thank the staff and participants of the Elsa Study for their important contributions.

## Author Contributions

**Conceptualization:** Maria de Fátima Haueisen Sander Diniz.

**Data curation:** Paulo A. Lotufo, Sandhi Maria Barreto.

**Formal analysis:** Maria de Fátima Haueisen Sander Diniz, Alline Maria Rezende Beleigoli, Isabela M. Benseñor, Sandhi Maria Barreto.

**Investigation:** Maria de Fátima Haueisen Sander Diniz.

**Methodology:** Maria de Fátima Haueisen Sander Diniz, Alline Maria Rezende Beleigoli, Isabela M. Benseñor, Paulo A. Lotufo, Alessandra C. Goulart, Sandhi Maria Barreto.

**Supervision:** Isabela M. Benseñor.

**Writing – original draft:** Maria de Fátima Haueisen Sander Diniz.

**Writing – review & editing:** Alline Maria Rezende Beleigoli, Isabela M. Benseñor, Paulo A. Lotufo, Alessandra C. Goulart, Sandhi Maria Barreto.

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
