## [Decision Letter · Decision Letter 0]

14 Nov 2019

PONE-D-19-24212

Association between TSH levels and adiposity markers at the baseline of the Brazilian Longitudinal Study of Adult Health (ELSA–Brasil)

PLOS ONE

Dear Mrs. Diniz,

Thank you for submitting your manuscript to PLOS ONE. After careful consideration, we feel that it has merit but does not fully meet PLOS ONE’s publication criteria as it currently stands. Therefore, we invite you to submit a revised version of the manuscript that addresses the points raised during the review process.

We would appreciate receiving your revised manuscript by Dec 29 2019 11:59PM. To enhance the reproducibility of your results, we recommend that if applicable you deposit your laboratory protocols in protocols.io, where a protocol can be assigned its own identifier (DOI) such that it can be cited independently in the future. For instructions see: http://journals.plos.org/plosone/s/submission-guidelines#loc-laboratory-protocols

We look forward to receiving your revised manuscript.

Kind regards,

Sabine Rohrmann

Academic Editor

PLOS ONE

Journal Requirements:

2. Please confirm whether the initial ethics committee approval for the ELSA-Brasil study also covers your cross-sectional study in the Methods section of your manuscript. If the ethics committee approval does not extend to this cross-sectional study, please include the full name of the IRB/ethics committees that reviewed and approved this study, including the name of the affiliated institution(s) if applicable. We additionally ask that you include your IRB/ethics committee approval number in your ethics statement.

"The study was supported by the Brazilian Ministries of Health (DECIT)

420 and of Science and Technology (FINEP/CNPq).".

 "Funding: The ELSA-Brasil study was supported by the Brazilian Ministry of Health(Science and Technology Department) and the Brazilian Ministry of Science and Technology (Financiadora de Estudos e Projetos and CNPq National Research Council) (grants 01 060010.00 RS, 01 06 0212.00 BA, 01 06 0300.00 ES, 01 06 0278.00 Mg, 01 06 0115.00 SP, 01 06 0071.00 RJ).".

Reviewers' comments:

Reviewer's Responses to Questions

**Comments to the Author**

1. Is the manuscript technically sound, and do the data support the conclusions?

Reviewer #1: Yes

Reviewer #2: Yes

2. Has the statistical analysis been performed appropriately and rigorously? 

Reviewer #1: Yes

Reviewer #2: Yes

3. Have the authors made all data underlying the findings in their manuscript fully available?

Reviewer #1: Yes

Reviewer #2: Yes

4. Is the manuscript presented in an intelligible fashion and written in standard English?

Reviewer #1: Yes

Reviewer #2: Yes

5. Review Comments to the Author

Reviewer #1: In this manuscript the authors investigated the association of TSH with markers of obesity in 11224 individuals.

Comments:

Title:

- Please remove the word longitudinal, since only cross-sectional data was considered.

- Mention that only TSH levels in the reference range were considered

Introduction:

Lines 59-62: Here some more studies should be included, e.g. the Asvold studies and PMID:27393002 ; please specify which studies adjusted for smoking.

Change "maximum reference value" to "upper reference limit".

Lines 65-66: I don't agree that the BMI is the best marker of overall obesity. Do you have a reference for this?

Material & Methods:

- When were blood samples taken? It is well known that TSH follows a circadian rhythm peaking in the morning which may have affect the results.

- Were all examiners certified for the examinations? Do you have data on intra- and inter observer variation?

- I suggest to not adjust for Diabtes and cardiovascular disease, since this may introduce a collider-stratification bias.

Results

Line 217: Change multivariate to multivariable

Table 2 needs reformatting - it is very hard to read. Please just state whether the p-value is below or above 0.05. Statistical tests are yes/no decisions.

- It seems that you used stepwise procedures on the potentially confounding variables (or what do you mean by final model?). This is no good procedure, because we are not in the world of prediction. The aim of this manuscript is to investigate whether TSH is associated with markers of body composition - thus, to draw causal inference. For this it is better to define the model a priori to data analysis based on a Directed Acyclic Graph (DAG) and afterwards all models are run with the same confounder set.

- Stratified analyses: Did you test whether the interaction terms of smoking or TPOab with logTSH were significantly associated with markers of body composition? If not significant, stratified analyses are not useful.

Reviewer #2: This is a large population based cross sectional study assessed the association between TSH levels and adiposity markers at the baseline of the Brazilian Longitudinal Study of Adult Health (ELSA–Brasil).

The authors found significant associations of SH with BMI and obesity and the association of TSH with waist circumference only among women. NC and steatosis were not related to TSH levels.

The study has strength of good design and methodology considering various confounders and large sample size and assessed 2 variables e.g. NC and steatosis which less have been studied previously. Although, the subject is not a new challenging issue especially within cross sectional design which can not show any cause and effect relationship and thyroid hormones especially T3/T4 ratio have not been assessed.

Following points should be addressed

Introduction is well written.

In methods CVs of laboratory measurements should be mentioned.

In Results Section Table 1 the sex-split results of the association of BMI with TSH is not reported, while for NC, WC and steatosis the results reported sex splitted and also adjusted foe sex in model 1!!!???? Please explain

If there is interaction between sex and TSH with BMI, all analysis should be sex splitted instead of adjusting for sex.

It is not clear all over the text (results and discussion), whether the association of TSH and BMI was assessed within 3 categories of BMI and this relationship was significant only in obese individuals or TSH was associated with obesity compared with other categories of normal and overweight ???

6. PLOS authors have the option to publish the peer review history of their article (what does this mean?). If published, this will include your full peer review and any attached files.

Reviewer #1: No

Reviewer #2: Yes: Ladan Mehran

---

## [Author Response · Author response to Decision Letter 0]

7 Jan 2020

Reviewer 1: Thank you for the valuable suggestions.

Here are the answers:

Title:

- Please remove the word longitudinal, since only cross-sectional data was considered.

- Mention that only TSH levels in the reference range were considered

- We changed the title for:

“Association between TSH levels in the reference range and adiposity markers at the baseline of the ELSA–Brasil study”

- The word longitudinal is related to the name of the cohort study “Brazilian Longitudinal Study of Adult Health” but, in fact, this analysis is cross-sectional, from the baseline data of the cohort. 

Introduction:

Lines 59-62: Here some more studies should be included, e.g. the Asvold studies and PMID:27393002 ; please specify which studies adjusted for smoking.

We updated the References list including other studies as Asvold BO et al. (JAMA Intern Med. 2015 Jun; 175(6): 1037–1047) and Tiller D et al. (Thyroid. 2016;26:1205-14). We also specified which studies adjusted or not for smoking

Line 59-60- However, among these studies, some adjusted for smoking status7-17 and others do not. 18-21 

Change "maximum reference value" to "upper reference limit".

We changed the word maximum by upper:

Line 60- “Also, the upper reference limit of TSH was not the same among the studies.7-21”

Lines 65-66: I don't agree that the BMI is the best marker of overall obesity. Do you have a reference for this?

- We mentioned that BMI is the best marker of overall obesity because this is the indirect measurement most commonly used and that recommended by the World Health Organization for the evaluation of overweight/obesity in adults. The World Health Organization stated that “ A crude population measure of obesity is the body mass index (BMI)” and “BMI provides the most useful population-level measure of overweight and obesity as it is the same for both sexes and for all ages of adults”.

24.World Health Organization. Obesity and overweight. Key facts. Available at: https://www.who.int/en/news-room/fact-sheets/detail/obesity-and- overweighthttps:// Accessed:11/25/2019.

- Also an Endocrine Society Scientific Statement of 2018 stated that “BMI provides the most useful population-level measurement of overweight and obesity, and numerous large population studies across multiple continents have demonstrated its utility as an estimate of risk.

25. Bray G, Heisel WE, Afshin A, Jensen MD, Dietz WH, Long M,et al. The Science of Obesity Management: An Endocrine Society Scientific Statement. Endocr Rev. 2018;39:79-132.

- We used the term “overall obesity” as assessed by BMI, in order to differentiate from abdominal obesity, assessed by waist circumference. 

26. Hu F. Measurements of adiposity and body composition. In: Hu FB, ed. Obesity Epidemiology. New York, United States: Oxford University Press; 2008:53-83. 

- We rephrased the Introduction and the Methods section in order to clarify the use of BMI as a marker of adiposity:

Line 71- Body mass index has been considered the best marker of overall obesity and is mainly used in large epidemiological studies.24-26 

Line 144: Since BMI is considered the most practical measure of overall obesity for populations24-26,35 , we used BMI as a proxy of overall obesity in this study.

Material & Methods:

- When were blood samples taken? It is well known that TSH follows a circadian rhythm peaking in the morning which may have affect the results.

We rephrased the Laboratory Measurements at Methods Section

Line 118 - Blood samples were collected after an overnight fast by “Blood samples were collected after a 12-hour fasting.30”

All the blood samples were collected with the same protocol, so do not impose bias to these analyses. We referenced studies where TSH was assessed in non-fasting blood samples (Asvold, Bjergved), but also others that used fasting samples (Kithara, Fox, Sakurai).

- Were all examiners certified for the examinations? Do you have data on intra- and inter observer variation?

All the anthropometric parameters were carried out using standard equipment and techniques by trained, certified professionals under rigorous quality control. The intra-class correlation coeficient for repeat measurements of waist circumference, an anthropometric measure sources of variability, was 0.98 (95% CI 0.85–1.0).

29.-Aquino S, Aquino EML, Barreto SM, Carvalho MS, Chor D, Duncan BB et al. ELSA-Brasil (Brazilian Longitudinal Study of Adult Health): objectives and design. Am J Epidemiol. 2012;175:315-24.

30.Schmidt MI, Griep RH, Passos VM, Luft VC, Goulart AC, Menezes GM de S et al. Strategies and development of quality assurance and control in the ELSA-Brasil. Rev Saúde Pública 2013; 47(Suppl 2): 105–112. 

- I suggest to not adjust for Diabetes and cardiovascular disease, since this may introduce a collider-stratification bias.

We decided to adjust for diabetes and cardiovascular because these conditions may impose lifestyle modifications that could contribute to anthropometric alterations. As suggested, we performed a new analysis without adjustments for cardiovascular disease or diabetes and final models had no significantly different results: for BMI, eβ=1.005[1.003-1.007] p<0.01; for categorical BMI, overweight category: eβ=1.018[0.997-1.039] p=0.09, obesity category: eβ=1.051[1.025-1.077] p<0.01; for waist circumference, among men: eβ=1.000[0.999-1.002] p=0.20, among women: eβ=1.002[1.0006-1.003] p=0.002; for neck circumference, among men: eβ=1.003[0.998-1.007] p=0.26, among women: eβ=1.005[0.999-1.009] p=0.08; for steatosis, mild: eβ=0.99[0.97-1.014] p=0.49, moderate/severe: eβ=1.007[0.95-1.069] p=0.83. 

Results

Line 217: Change multivariate to multivariable

- We reworded the text as suggested “In multivariable linear regression models” (Line 234)

Table 2 needs reformatting - it is very hard to read. Please just state whether the p-value is below or above 0.05. Statistical tests are yes/no decisions.

-As suggested, we reformatted Table 2 and 3, including only crude and final models, and stated p value as below or above 0.05. The objective of displaying both at the same table was to demonstrate that the adjustments had small effects on the results. 

- It seems that you used stepwise procedures on the potentially confounding variables (or what do you mean by final model?). This is no good procedure, because we are not in the world of prediction. The aim of this manuscript is to investigate whether TSH is associated with markers of body composition - thus, to draw causal inference. For this it is better to define the model a priori to data analysis based on a Directed Acyclic Graph (DAG) and afterwards all models are run with the same confounder set.

As the study had a cross-sectional design, the objective was not to draw causal inference. In fact, we plan this for a future analysis of longitudinal data. This theme is still a challenge, even on longitudinal studies, where some authors have demonstrated causal relationship between TSH and BMI/waist circumference and others have not. 

We pointed out the confounding variables, and the analysis was drawn not to compare models, but rather to group the variables as socio-demographic, comorbidities, health related conditions. In fact, it is not a stepwise model, and we did not try to build an algorithm. The analysis had an epidemiological basis, with the adjustment for the potential confounders.

In order to clarify, we rephrased the Statistical analysis: “All models were adjusted for the following covariates: Model 1- age (years) and sex (only for BMI and steatosis); Model 2- Model 1 plus race/skin color (white, black or mixed), education, TPOAb (negative/positive), diabetes (yes/no), cardiovascular disease (yes/no), and glomerular filtration rate; and Model 3 – Model 2 plus lifestyle behaviors (smoking, alcohol use, physical activity)” – Line 195-199.

- Stratified analyses: Did you test whether the interaction terms of smoking or TPOab with logTSH were significantly associated with markers of body composition? If not significant, stratified analyses are not useful.

Thank you for this important suggestion. We have tested the interaction terms of smoking and TPOAb with log TSH. The results of the interaction terms were not significant for BMI, BMI categorized. However, the interaction term of smoking and waist among women, and smoking and steatosis were borderline: p=0.08 for waist and p=0.06 for moderate/severe steatosis, suggesting a potential impact of these interactions. The interaction term of TPOAb and neck was significant (p=0.03) only among men. The interaction term of TPOAb and steatosis moderate/severe was also significant (p=0.01). We changed the Methods section and presented the results after the stratified analysis only when it was appropriate. 

- Reviewer 2- Thank you for the valuable suggestions. Here are the answers:

- In methods CVs of laboratory measurements should be mentioned.

Thank you for this relevant suggestion.

Regarding the CVs of laboratory measurements, we included the reference 33: the following manuscript describes all of them.

33.LADWIG, R. et al. Variability in baseline laboratory measurements of the Brazilian Longitudinal Study of Adult Health (ELSA-Brasil). Braz J Med Biol Res [online]. 2016, vol.49, n.9, e5381. Epub Aug 01, 2016. ISSN 1414-431X. 

We rewrote the manuscript including the information of CV: “Intra-assay coefficients of variation (CV) of the TSH, HDL cholesterol, triglycerides and A1c were, respectively, 3.32%, 2.0%,1.6%, 0.86%. Inter-assay CV for lower and higher control of TSH were 4.65 and 5.89, respectively” (Lines 133-136).

In Results Section Table 1 the sex-split results of the association of BMI with TSH is not reported, while for NC, WC and steatosis the results reported sex splitted and also adjusted foe sex in model 1!!!???? Please explain

If there is interaction between sex and TSH with BMI, all analysis should be sex splitted instead of adjusting for sex.

We did not show the results of BMI, categorical BMI and steatosis by sex, because the there was no evidence that sex modified the association between these adiposity markers and TSH levels through the analyses of interaction terms. 

It is not clear all over the text (results and discussion), whether the association of TSH and BMI was assessed within 3 categories of BMI and this relationship was significant only in obese individuals or TSH was associated with obesity compared with other categories of normal and overweight ???

Thank you for the comment. We stated at the ‘Methods’ section that BMI was analyzed as a continuous variable, and also as a categorical variable: categorized as 18.5-25.0 kg/m2 (reference), ≥ 25- 29.9 kg/m2, and ≥ 30 kg/m2. The linear model with categorical BMI compared the reference group (normal weight) with overweight and with obesity, as depicted in Table 3. We tried to clarify this at the discussion:

Line 302-304- “In the analysis with BMI categories, this association was present only among participants with a BMI≥ 30 kg/m2 in comparison with normal weight category.”

---

## [Decision Letter · Decision Letter 1]

24 Jan 2020

Association between TSH levels within the reference range and adiposity markers at the baseline of the ELSA–Brasil study

PONE-D-19-24212R1

Dear Dr. Diniz,

We are pleased to inform you that your manuscript has been judged scientifically suitable for publication and will be formally accepted for publication once it complies with all outstanding technical requirements.

With kind regards,

Sabine Rohrmann

Academic Editor

PLOS ONE

Additional Editor Comments (optional):

Reviewers' comments:

Reviewer's Responses to Questions

**Comments to the Author**

1. If the authors have adequately addressed your comments raised in a previous round of review and you feel that this manuscript is now acceptable for publication, you may indicate that here to bypass the “Comments to the Author” section, enter your conflict of interest statement in the “Confidential to Editor” section, and submit your "Accept" recommendation.

Reviewer #1: All comments have been addressed

2. Is the manuscript technically sound, and do the data support the conclusions?

Reviewer #1: Yes

3. Has the statistical analysis been performed appropriately and rigorously? 

Reviewer #1: Yes

4. Have the authors made all data underlying the findings in their manuscript fully available?

Reviewer #1: No

5. Is the manuscript presented in an intelligible fashion and written in standard English?

Reviewer #1: Yes

6. Review Comments to the Author

Reviewer #1: Thank you for adressing all my questions and suggestions. I have no further comments.

7. PLOS authors have the option to publish the peer review history of their article (what does this mean?). If published, this will include your full peer review and any attached files.

Reviewer #1: No

---

## [Editor Report · Acceptance letter]

27 Jan 2020

PONE-D-19-24212R1 

Association between TSH levels within the reference range and adiposity markers at the baseline of the ELSA–Brasil study 

Dear Dr. Diniz:

I am pleased to inform you that your manuscript has been deemed suitable for publication in PLOS ONE. Congratulations! Your manuscript is now with our production department. 

With kind regards,

on behalf of

Dr. Sabine Rohrmann 

Academic Editor

PLOS ONE